# Novel Pituitary Actions of TAC4 Gene Products in Teleost

**DOI:** 10.3390/ijms222312893

**Published:** 2021-11-29

**Authors:** Xuetao Shi, Cheng Ye, Xiangfeng Qin, Lingling Zhou, Chuanhui Xia, Tianyi Cai, Yunyi Xie, Zhan Yin, Guangfu Hu

**Affiliations:** 1College of Fisheries, Huazhong Agricultural University, Wuhan 430070, China; sxt0902@163.com (X.S.); u3007072@connect.hku.hk (C.Y.); qinxf@connect.hku.hk (X.Q.); llz9872@163.com (L.Z.); xiachuanhui@webmail.hzau.edu.cn (C.X.); caity13@163.com (T.C.); caroline_xieyunyi@163.com (Y.X.); 2State Key Laboratory of Freshwater Ecology and Biotechnology, Institute of Hydrobiology, Chinese Academy of Sciences, Wuhan 430072, China

**Keywords:** tac4, hemokinin, neurokinin receptor, pituitary, teleost

## Abstract

Tachykinin 4 (TAC4) is the latest member of the tachykinin family involved in several physiological functions in mammals. However, little information is available about TAC4 in teleost. In the present study, we firstly isolated TAC4 and six neurokinin receptors (NKRs) from grass carp brain and pituitary. Sequence analysis showed that grass carp TAC4 could encode two mature peptides (namely hemokinin 1 (HK1) and hemokinin 2 (HK2)), in which HK2 retained the typical FXGLM motif in C-terminal of tachyinin, while HK1 contained a mutant VFGLM motif. The ligand-receptor selectivity showed that HK2 could activate all 6 NKRs but with the highest activity for the neurokinin receptor 2 (NK2R). Interestingly, HK1 displayed a very weak activation for each NKR isoform. In grass carp pituitary cells, HK2 could induce prolactin (PRL), somatolactin α (SLα), urotensin 1 (UTS1), neuromedin-B 1 (NMB1), cocaine- and amphetamine-regulated transcript 2 (CART2) mRNA expression mediated by NK2R and neurokinin receptor 3 (NK3R) via activation cyclic adenosine monophosphate (cAMP)/protein kinase A (PKA), phospholipase C (PLC)/inositol 1,4,5-triphosphate (IP3)/protein kinase C (PKC) and calcium^2+^ (Ca^2+^)/calmodulin (CaM)/calmodulin kinase-II (CaMK II) cascades. However, the corresponding stimulatory effects triggered by HK1 were found to be notably weaker. Furthermore, based on the structural base for HK1, our data suggested that a phenylalanine (F) to valine (V) substitution in the signature motif of HK1 might have contributed to its weak agonistic actions on NKRs and pituitary genes regulation.

## 1. Introduction

In mammals, the tachykinin (TAC) family includes three members, namely tachykinin 1 (TAC1), tachykinin 3 (TAC3) (tachykinin 2 (TAC2) in rodents), and TAC4, respectively. Among them, TAC1 gene encodes substance P (SP) and neurokinin A (NKA). SP is known to be involved in the regulation of pain control/injury [1], neurogenicinflammation [2] and obesity [3] with mammals. Compared with mammals, teleost have undergone an additional genome duplication during evolution, which is called fish-specific genome duplication (FSGD) or the 3R hypothesis [4]. So there were two isoforms TAC1 (namely tachykinin 1 isoforms a (TAC1a) and tachykinin 1 isoforms b (TAC1b)) and TAC3 (namely tachykinin 3 isoforms a (TAC3a), and tachykinin 3 isoforms b (TAC3b)), respectively. Similar to mammals, teleost TAC1 also encodes two mature peptides, namely SP and NKA, respectively [5]. In teleost, SP and NKA can trigger the secretion of luteinizing hormone β (LHβ), PRL, and SLα in the pituitary cells of carp, and the transcriptional levels of PRL and SLα increase in parallel. Short-term SP treatment (3 h) induced the release of LHβ, while prolonged induction time (24 h) inhibited LHβ mRNA expression [5].

In mammals, TAC3 encodes neurokinin B (NKB), is known to be involved in the regulation of smooth muscles of the gastrointestinal tract, secretion of intestinal epithelial fluid, vasodilation, and stimulating sperm motility [6,7,8]. Recent studies have found that NKB is a key regulator of mammalian reproductive function, especially controlling the release of gonadotropin-releasing hormone (GnRH) in the hypothalamus [9,10]. In contrast to mammals, TAC3 does not only encode NKB, but also a novel tachykinin peptide in teleost, named neurokinin B-related peptide (NKBRP) [11]. The research on the function of the TAC3 gene products in fish has mainly focused on the regulation of the reproductive process. Previous studies have found that intraperitoneal injection of NKB or NKBRP could both induce LHβ secretion in sexually mature female zebrafish [12]. In addition, our previous studies have shown that both NKB and NKBRP can promote PRL and SLα secretion and mRNA synthesis in grass carp pituitary cells [11].

Compared to TAC1 and TAC3, TAC4 is the last member of the tachykinin family. In 2000, Zhang et al. was first isolated the TAC4 gene from mouse hematopoietic stem cells, and named its gene production as hemokin 1 (HK1) [13]. Subsequently, TAC4 was cloned from rats and humans. Among them, the mouse hemokinin 1 (mHK1) encoded by the rat TAC4 gene has extremely high similarity with rat hemokinin 1 (rHK1), while the human race TAC4 gene encoding product was fairly different [14]. The TAC4 gene can encode six neuropeptides, namely HK1, HK1 (4–11), endokinin A (EKA), endokinin B (EKB), endokinin C (EKC), and endokinin D (EKD), respectively [15,16,17]. In mammals, recent studies have shown that HK1 and EKs were new mediators of pain and inflammation, and also play the crucial role in the hematopoietic system, anti-anxiety, and anti-depression [18,19,20]. However, little information is available about TAC4 in teleost.

In the present study, using grass carp (*Ctenopharyngodon idellus*) as a model, we try to examine the pituitary actions of TAC4 gene product in teleost. Firstly, the grass carp TAC4 were cloned, which encoded two mature peptides, namely HK-1 and HK-2, respectively. Secondly, six potential NKRs were isolated from grass carp. Then, by using transfection and dual-luciferase detection, we tried to confirm the specific receptor for HK-1 and HK-2, respectively. Thirdly, using grass carp pituitary cells as a model, we try to examine the direct pituitary actions of HK-1 and HK-2 in teleost. Finally, we try to clarify the mechanism of functional differences between HK-1 and HK-2 in teleost. 

## 2. Materials and Methods

### 2.1. Animals and Chemicals

Two-year-old grass carps with a body weight of 1.7 ± 0.2 kg were purchased from a local market and kept in a well-aerated 250 L aquaria under a 12 h light/12 h dark photoperiod at 28 ± 1 °C for seven days. Grass carps at this stage were prepuberal and the sex character was not obvious, so breed fish with mixed sexes for pituitary cell preparation. All experimental procedures were approved by the Huazhong Agricultural University for Laboratory Animal Care (Ethical Approval No. HBAC20091138; Date: 15 November 2009). Grass carps selected in the experiment were all subjected to complete anesthesia at 0.05% MS222 (Sigma, St. Louis, MO, USA). Grass carp HK2 (QKFQT**F**VGLM-NH_2_), HK1 (TPGLQQ**V**FGLM-NH_2_), and HK1-V7F (TPGLQQ**F**FGLM-NH_2_) were synthesized by GeneScript Corporation (Piscataway, NJ, USA). The three mature peptides were subpackaged at 1 mM concentration using ultrapure water and stored at −80 °C.

### 2.2. Molecular Cloning and Tissue Distribution of Grass Carp TAC4 and NKRs

In order to clarify the distribution of TAC4 and its receptors in various tissues of grass carp, total RNA were extracted from each brain subregion, pituitary gland, and peripheral tissues of grass carp. Then the concentration was detected by the Nanodrop 2000, and each tissue took an equal amount of total RNA for reverse transcription to prepare a cDNA template. Primer Premier 6 was used to design specific quantitative primers for these genes (Appendix A), and their transcript level in grass carp brain subregions and peripheral tissues was detected by real-time quantitative PCR (RT-qPCR). In these studies, β-actin was performed as an internal control. Sequence alignment and phylogenetic analysis of TAC4 and NKRs was conducted with ClustalX 2.1 and MEGA 7.0 (http://www.megasoftware.net/). The 3D structures for TAC4 and NKRs were predicted by I-TASSE (https://zhanglab.ccmb.med.umich.edu/I-TASSER/) and SWISS-MODEL (https://swissmodel.expasy.org/), respectively.

### 2.3. Functional Expression of Grass Carp NKRs in HEK293T Cells

To clarify the receptor selectivity of TAC4 gene products, a transfection experiment was conducted in HEK293T cells, and the fluorescent signal was detected by the firefly luciferase reporter gene detection kit. Firstly, the cDNA sequences of neurokinin receptor 1 isoforms a (NK1Ra), neurokinin receptor 1 isoforms b (NK1Rb), NK2R, neurokinin receptor 3 isoforms a1 (NK3Ra1), neurokinin receptor 3 isoforms a2 (NK3Ra2), and neurokinin receptor 3 isoforms b (NK3Rb) were isolated from grass carp pituitary, and the coding region of the six receptors were then subcloned into the eukaryotic expression vector pcDNA3.1. Secondly, the HEK293T cells were co-transfected with the plasmids of NFAT-Luc (Nuclear factor of activated T cells-luciferase), green fluorescent protein (GFP), and each pcDNA3.1-NKRs by using Lipofectamine 3000 (Thermo Fisher). The transfected HEK293T cells were incubated with various concentration HK-1 (1–10,000 nM), HK-2 (1–10,000 nM), or HK1-V7F (1–10,000 nM) for 24 h, respectively. Finally, the transfected cells were collected to detect the luciferase value according to the instructions of the firefly luciferase reporter gene detection kit.

### 2.4. Transcriptome Sequence and Bioinformatics

Grass carp pituitary cells were obtained by trypsin digestion [21]. Then, the pituitary cells were seeded into a 24-well cell culture plate at a density of 2.5 × 10^6^ cells/well, and incubated at 28 °C under 5% CO_2_ in cell culture incubator. The primary culture pituitary cells were treated by HK2 (1 μM) for 24 h. Then, the cells were collected to extract the total RNA by the Trizol method. The high-quality samples were sent to the Majorbio Genome Center (Shanghai, China) for transcriptomic sequencing. Using the negative binomial distribution model, the edgeR software was applied to calculate the differential expression based on the FPKM (fragments per kilobase of exon per million fragments mapped) value of the gene. The screening criteria for differentially expressed genes (DEGs) are FDR (false discovery rate) ≤ 0.05 and FC (fold change) ≥ 1.5. Functional annotation of gene ontology (GO) terms was analyzed by using Goatolls (http://github.com/tanghaibao/GOatolls), and KEGG functional classification were analyzed by using KOBAS (http://kobas.cbi.pku.edu.cn/home.do).

### 2.5. Real-Time Quantitative PCR (RT-qPCR) Validation

To further confirm the pituitary actions of HKs in grass carp, trypsin digestion method was used to prepare for grass carp primary cultured pituitary cells. In the time-dependent experiment, the pituitary cells were incubated with HK1 (1 µM) or HK2 (1 µM) for 3 h, 12 h, 24 h, and 48 h, respectively. In the dose-dependent experiment, the pituitary cells were incubated with HK1 (0.1–1000 nM) or HK2 (0.1–1000 nM) for 48 h, respectively. After drug treatment, total RNA was extracted from adherent cells by Trizol, and then subjected to reverse transcription using HifairTM III 1st Strand cDNA Synthesis Kit (Yeasen, Shanghai, China). In the present study, β-actin was used as the internal reference gene, and then, the transcript levels of PRL, SLα, CART2, UTS1, and NMB1 were detected by using a ABI 7500 real-time PCR system (see Appendix A for primer sequences and PCR condition). In addition, the plasmid DNA containing the gene coding sequence was serially diluted as a standard for data calibration.

### 2.6. Statistical Analysis

In this study, RT-qPCR was used to detect mRNA expression for TAC4, NKRs, PRL, SLα, CART2, UTS1, SN2, and NMB1. In the ABI7500 software, the dynamic mRNA concentration detection range of the double standard curve is 10^5^, and the correlation coefficient is >0.95. Since the transcript level of β-actin showed no significant changes, the data of PRL, SLα, CART2, UTS1, and NMB1 were converted into average percentages in the control group (% Ctrl). In the transfection experiment, the detected firefly luciferase activity was routinely normalized with the GFP activity expressed in the same well, expressed as the luciferin activity ratio Luc ratio. The data from four parallel experiments was presented as mean ± SEM in the figure. All the data were analyzed by ANOVA, and then subjected to Dunnett’s test in GraphPad Prism 7.0. The significant differences between groups were indicated by *p* < 0.05.

## 3. Result

### 3.1. Sequence Analysis and Tissue Distribution of TAC4 and NKRs

To examine the pituitary action of TAC4 gene products in grass carp, the cDNA sequence of TAC4 was firstly isolated from the brain of grass carp. Sequence analysis showed that grass carp TAC4 encoded two mature peptides, including an 11-a.a HK2 carrying the classical tachykinin signature motif (**F**XGLM) and a 10-a.a HK1 containing a mutated form of tacykinin consensus motif form (**V**FGLM) (Figure 1A,C). In addition, similar to the 3D structure of other tachykinin peptids, grass carp HK1 and HK2 were also found to be in the form of a short peptide with a α helix covering the region of the central core to the C-terminus (Figure 1D). Phylogenetic analysis revealed that the newly isolated grass carp TAC4 could be clustered in the clade of fish TAC4 with a close evolutionary relationship with common carp TAC4 (Figure 1E), and TAC4 group displayed a closer evolutionship with the TAC1 cluster compared to TAC3 (Figure 1E). Furthermore, we also isolated six neurokinin receptors from grass carp brain and pituitary, namely NK1Ra (Appendix A), NK1Rb (Appendix A), NK2R (Appendix A), NK3Ra1 (Appendix A), NK3Ra2 (Appendix A) and NK3Rb (Appendix A), respectively. Similar to mammals, the six NKRs in grass carp were all typical G protein- coupled receptor (Appendix A). Phylogenetic analysis showed that the 6 NKRs were clustered into three groups, including NK1R, NK2R, and NK3R group, respectively (Figure 1F). Finally, tissue distribution revealed that TAC4 was widely distributed in various brain regions except for the cerebellum, and its transcript level was extremely high in the medulla oblongata and the olfactory bulb (Figure 1B). In addition, six NKRs could all be widely detected in various brain subregions and pituitary, and NK1Rb was more highly detected in the brain compared to other NKRs (Figure 1B).

### 3.2. Receptor Selectivity of TAC4 Gene Products 

To clarify the receptor selectivity of HK1 and HK2 for the six NKRs, HEK-293T cell lines were established to stably express NK1Ra, NK1Rb, NK2R, NK3Rb, NK3Ra1, or NK3Ra2 were established, respectively. The cells were then used for transfection study with NFAT-Luc vectors, which was allowed for functional evaluation of the activation status of Ca^2+^-dependent pathway. The results showed that the HK2 could activate all 6 NKR isoforms, but HK1 displayed very weak activation for these NKRs (Figure 2). In the case of HK2, it exhibited affinity intensity was NK2R (EC_50_, 122.9 nM) > NK3Rb (EC_50_,145.6 nM) > NK1Ra (EC_50_,76.29 nM) ≈ NK3Ra1 (EC50,1344 nM) ≈ NK1Rb (EC50, 337.8 nM) > NK3Ra2 (EC50, 1232 nM) (Figure 2).

### 3.3. Transcriptomic Analysis of HK2 in the Pituitary

In order to reveal the pituitary actions of HK2 in lower vertebrates, the pituitary cells of grass carp were incubated with HK2 (1 μM) for 24 h. Then, high-throughput transcriptome technology was used to detect the changes in gene transcription levels in the pituitary after HK2 treatment (Figure 3). Compared with the control group, a total of 1151 differentially expressed genes (DEGs) were found in the HK2 treatment group, including 364 up-regulated DEGs and 787 down-regulated DEGs. The results of GO enrichment analysis showed that the up-regulated DEGs in the molecular function (MF) category were mainly involved in sequence-specific DNA binding, hormone activity, and receptor ligand activity, respectively (Figure 3A). In the biological process (BP) category, the main clusters were the negative regulation of metabolic processes, the G protein-coupled receptor signaling pathway, and the negative regulation of appetite (Figure 3A). Most of the cell components (CC) categories were enriched in kinetochore, plasma membrane, and cell junctions (Figure 3A). Down-regulated DEGs in the molecular function (MF) category were mainly clustered in G protein-coupled receptor activity, calcium ion binding, and steroid hormone receptor activity (Figure 3A). In the biological process (BP) category, they were mainly involved in cell adhesion, the regulation of innate immune response, and hormone-mediated signaling pathways (Figure 3A). In the cell components (CC) category, they were mostly enriched in the plasma membrane, ion channel complexes and transcription factor complexes (Figure 3A). The results of KEGG enrichment analysis showed that the up-regulated DEGs were mainly involved in the synthesis of steroid hormones, the cGMP-PKG signaling pathway, and the Jak-STAT signaling pathway. The down-regulated DEGs were mostly enriched in the cAMP signaling pathway, neuroactive ligand-receptor interactions, and phospholipase D signal pathway, respectively (Figure 3B). Finally, the GO enrichment group involved in reproduction function, growth factor, hormone activity, immune response, and feeding regulation are shown in Figure 4. In addition, the top 20 up-regulated DEGs are shown in Table 1, including CART2, PRL, SLα, UTS1, and NMB1. The top 20 down-regulated DEGs were displayed in Table 2, including somatostatin receptor type 2 isoforms a (SSTR2a), somatostatin receptor type 3 (SSTR3), opioid growth factor receptor-like protein 1 (OGFR), growth hormone secretagogue receptor type 1 (GHSR1), and fibroblast growth factor receptor 2 (FGFR2).

### 3.4. Regulation of HK1 and HK2 on the Key DEGs Expression in Grass Carp Pituitary Cells 

To further confirm the pituitary actions of HK1 and HK2 in grass carp, HK1 (1 µM) or HK2 (1 µM) were used to incubate grass carp pituitary cells for 3 h, 6 h, 24 h, and 48 h, respectively. The results showed that HK2 (1 µM) could significantly induce SLα (Figure 5A), PRL (Figure 5B), CART2 (Figure 6A), NMB1 (Figure 6B), and UTS1 (Figure 6C) mRNA expression in a time-dependent manner. However, HK1 (1 μM) showed no effect on these genes in grass carp pituitary cells (Figure 5 and Figure 6). To further evaluate the dose-dependence of these stimulatory effects on pituitary hormones (SLα and PRL) and feeding peptides (CART2, NMB1 and UTS1), grass carp pituitary cells were exposed to increasing concentrations (0.1–1000 nM) of HK1 or HK2 for 24 h, respectively. In this case, HK2 treatment could consistently induce SLα (Figure 5A), PRL (Figure 5B), CART2 (Figure 6A), NMB1 (Figure 6B), and UTS1 (Figure 6C) mRNA expression in a concentration-related fashion. Similar to our time-course study, increasing levels of HK1 were ineffective in altering transcript expression of the five genes mRNA expression in grass carp pituitary cell even up to 1 µM concentration (Figure 5 and Figure 6). 

### 3.5. Receptor Specificity and Post-Receptor Signal Transduction for HK2-Induced SLα and PRL mRNA Expression in Grass Carp Pituitary Cells 

In this experiment, a pharmacological approach was used to clarify the receptor specificity for SLα and PRL regulation by HK2. Pituitary cells were incubated for 24 h with grass carp HK2 (1 μM) with simultaneous treatment of the NK1R antagonist Rolaptiant (10 μM) or NK3R antagonist SB222200 (10 μM), respectively. Similar to the results of the proceeding studies, HK2 could increase the basal transcript level of SLα and PRL. The stimulatory effects on SLα mRNA expression could be totally blocked by co-treatment with the NK3R anatagonist SB222200, but NK1R antagonist could only partially block HK2-induced SLα mRNA expression (Figure 7A). In the parallel experiment, HK2-induced PRL mRNA expression could not be totally abolished by NK1R antagonist Rolaptiant or NK3R antagonist SB222200, respectively (Figure 7A). 

To elucidate the signal transduction for PRL and SLα regulation by HK2, the possible involvement of the cAMP-dependent cascade was examined by using the inhibitors for the cAMP pathway. As shown in Figure 7B, co-treatment with AC inhibitor MDL1230A (10 μM) or PKA inhibitor H89 (10 μM) were both effective in blocking the stimulatory effects of HK2 (1 μM) on PRL and SLα mRNA expression. In parallel experiments for testing the functional role of the PLC/IP3/PKC pathway, HK2-induced PRL and SLα mRNA expression could be both totally abolished by simultaneous treatment with the PLC inhibitor U73122 (10 μM), PKC inhibitor GF109203X (10 μM), or IP3 receptor blocker 2-APB (10 μM), respectively (Figure 7C). Furthermore, HK2-induced PRL and SLα mRNA expression were found to be suppressed or totally abolished by co-treatment with the VSCC inhibitor nifedipine (10 μM), CaM antagonist calmidazolium (1 μM) or CaMK-II blocker KN62 (10 μM), respectively (Figure 7D). 

### 3.6. Receptor Specificity and Signal Transduction for HK2-Induced UTS1, NMB1 and CART2 mRNA Expression

In this part, the pharmacological approach was also recruited to clarify the receptor specificity and signal transduction for CART2, UTS1, or NMB1 transcript regulation by HK2. As shown in Figure 8A, the stimulatory effects of HK2 on CART2, UTS1, or NMB1 mRNA expression could be totally abolished by NK3R antagonist SB222200 (10 μM) but not with the NK1R antagonist Rolaptiant (10 μM) (Figure 8A). To shed light on the signal transduction for CART2, UTS1, or NMB1 transcript regulation by HK2, the possible involvement of cAMP-dependent pathway was examined firstly using the co-treatment with various inhibitors for the individual components of cAMP pathway. As shown in Figure 8B, HK2-induced CART2, UTS1, or NMB1 mRNA expression could be abolished by simultaneous incubation with the AC inhibitor MDL12330A (10 μM) or PKA inhibitor H89 (10 μM). In the parallel studies, co-treatment with PLC inhibitor U73122 (10 μM), PKC inhibitor GF109203X (10 μM), or IP3 receptor blocker 2-APB (10 μM) were all effective in blocking the stimulatory effects of HK2 (1 μM) on CART2, UTS1 or NMB1 mRNA expression (Figure 8C). Furthermore, HK2-induced CART2, UTS1, or NMB1 mRNA expression were found to be suppressed/totally abolished by co-treatment with VSCC inhibitor nifedipine (10 μM), CaM antagonist calmidazolium (1 μM) or CaMK-II blocker KN62 (10 μM), respectively (Figure 8D).

### 3.7. Phe to Val Mutation in FXGLM Motif in UTS1, PRL and SLα Regulation by HK2

As carp HK1 is unique for having a Phe to Val mutation in the “**F**XGLM” signature motif, the biological relevance of this mutation in pituitary hormone regulation was examined by mutating the Val residue at position 7 of carp HK1 back to Phe as in the case of the typical “**F**XGLM” motif. As shown in Figure 9A, the dose-response study has shown that the HK1-V7F mutant could significantly enhance the stimulatory effects of HK1 on SLα, PRL, and UTS1 mRNA expression to levels comparable to that induced by HK2. In addition, Phe to Val mutation in the “**F**XGLM” motif in HK1 on binding affinity to NKR were also examined. In HEK-293T cells with stable expression grass carp each NKR, co-transfection with NFAT-Luc expressing vectors were conducted to allow for biochemical probing of functional activation of NKR. Similar to the results of pituitary hormone genes regulation in carp pituitary cells, carp HK1 with a “**V**XGLM” motif was found to be a relatively low efficient stimulant for six NKR isoforms (Figure 2). In parallel experiments with the treatment of increasing doses of the Val to Phe mutant HK1-V7F, the stimulatory effects on NFAT-Luc mediated luciferase activity expression was notably enhanced with significant increases in the corresponding ED_50_ value from >10 μM to the nanomolar dose range (Figure 2). 

Furthermore, to verify whether HK1 could function as an endogenous antagonist of NKRs, we examined the antagonist effect of HK1 on HK2-induced pituitary hormone genes expression. As shown in Figure 9B, a 24 h static incubation with a 0.1 µM dose carp HK2 was effective in triggering significant rises in SLα, PRL, and UTS1 mRNA expression in grass carp pituitary cells. Of note, the stimulatory effects of HK1 treatment were found to be much weaker compared to the corresponding stimulation induced by HK2 (Figure 9B). In the same experiment, co-treated with increasing levels of carp HK1 (1 µM and 10 µM doses) could not suppress the stimulatory actions caused by HK2 on SLα, PRL, and UTS1 mRNA expression (Figure 9B). 

## 4. Discussion

In mammals, multiple tachykinin genes with different gene products, including TAC1 encoding SP and NKA, TAC3 (also referred to as TAC2 in rodents) encoding NKB and TAC4 encoding HK1 and EKs, have been widely reported [22]. Compared to mammals, due to the third round of whole-genome duplication, two TAC1 isoforms (TAC1a and TAC1b) [5] and two TAC3 isoforms (TAC3a and TAC3b) [23] have been reported in teleost. However, in the present study, only one TAC4 isoform was isolated in teleost, which might be the result of the non-functionalization by forming pseudogenes or deletion/mutations leading to the loss of redundant genes [24] (Appendix A). In mammals, TAC4 gene products were various in different species. TAC4 in mice and rats can encode one hemokinin mature peptide with 11 amino acid length [25]. In human, TAC4 encodes for hemokinin-1 (hHK-1), but its sequence is different from its murine counterpart. It is spliced into four alternative transcripts (α, β, γ and δ) that give rise to four different peptides which have been named endokinis A (EKA), B (EKB), C (EKC), and D (EKD), respectively [26,27]. In the present study, in contrast to mammals, TAC4 in teleost could not only encode HK-1, but also another mature tachykinin HK-2, which might be due to a loss of segmentally duplicated gene fragments in TAC4 during tetrapod evolution. 

In mammals, the biological action of tachykinins is mediated by three GPCRs, namely NK1R, NK2R, and NK3R, which are stimulated preferentially by SP, NKA, and NKB, respectively [28]. Both in mice and rats, HK-1 has a similar affinity to SP at the NK1R [29,30], while human HK-1 binds to NK1R with a much lower affinity than SP [31]. In addition, high-dose human HK-1 can also bind to NK2R and NK3R [14]. In the present study, due to the third round whole-genome duplication, 6 NKRs were detected in grass carp, namely NK1Ra, NK1Rb, NK2R, NK3Ra1, NK3Ra2, and NK3Rb, respectively. Based on the ligand-receptor selectivity, we found that HK2 could activate all 6 NKR isoforms, but showed the highest affinity for NK2R. Our previous study has revealed that carp NK2R is a multiligand receptor, which could be activated by SP, NKA, NKBa, NKBRPa, NKBRPb with comparable efficacy and potency [5]. These results indicated that teleost hemokinin preferentially stimulated the multiligand receptor NK2R, which suggests that hemokinin in teleost may analog the function of other tachykinins through activation of NK2R. 

In mammals, many studies have focused on the actions of hemokinin on immunological regulation and inflammation [32]. However, little information is available about its endocrinology function. In the present study, firstly, we found that six NKRs could be detected in grass carp pituitary, which indicated that HKs play an important role in the pituitary. Then, using grass carp pituitary cells as a model, we found that carp HK2 could significantly induce PRL and SLα mRNA expression mediated by NK2R and NK3R via activation of the cAMP/PKA, PLC/IP3/PKC, and Ca^2+^/CaM/CaMK II cascade. Similar results were also reported in our previous TAC3 studies, which indicated that NKB could significantly induce pituitary PRL and SLα synthesis and secretion [17]. In fish models, PRL is involved in a wide range of physiological functions, including reproductive migration [33], gonadal maturation [34], reproductive cycling [35,36], nesting and brooding behaviors [37], and osmoregulation [38]. As the latest member of the growth hormone (GH)/PRL family, somatolactin was a pituitary hormone unique to fish species and has been reported to play a functional role in the regulation of reproduction [39], growth [40] and pigmentation [41]. These results, as a whole, suggested that HK2 was involved in pituitary regulation of PRL and SLα mRNA expression, which presumably may contribute to the reproduction and growth metabolism in teleost.

In addition to the regulation of pituitary PRL and SLα, our present study also found that HK2 could significantly induce pituitary UTS1, CART2, and NMB1 mRNA expression mediated by NK3R via activation of cAMP/PKA, PLC/IP3/PKC, and Ca^2+^/CaM/CaMK II signal pathways. Interestingly, our recent study also found that NKB could stimulate UTS1, CART2, and NMB1 mRNA expression in grass carp pituitary cells [42]. Previous studies reported that CART could inhibit feeding in rats [43] and goldfish [44]. Similarly, NMB has also been found to inhibit feeding and gastric peristalsis [45,46]. Fish UTS1 showed a closed structural and biological homology with the corticortropin-releasing hormone (CRH) family. In mammals, CRH could reduce food intake and induce energy expenditure [47]. In teleost, the CRH system could also exert an anorexigenic effect [48]. These results, taken together, suggested that HK2 might be involved in the regulation of feeding in teleost.

In our functional studies with carp pituitary cells, interestingly, the potency and efficacy of the other TAC4 gene product, namely HK1, was found to be ineffective for UTS1, PRL, and SLα mRNA expression. In addition, HK1 could barely activate any NKR isoforms in HEK-293T cells. Sequence analysis showed that HK1 from different fish species was also very unique in having a well-conserved Phe to Val mutation in the signature motif “VXGLM” located in the C-terminal. The functional role of the new signature motif “VXGLM” in HK1 was currently unclear, but an “FXGLL” motif has been previously reported in human EKC and EKD despite the fact that the other gene products of human TAC4, namely EKA and EKB, still have the original “FXGLM” signature sequence [49]. In human patients with TAC3 gene mutations, a Thr for Met mutation in the “FXGLM” motif of NKB was known to cause hypogonadism or even infertility in adulthood [1]. To confirm that the low intrinsic activity of HK1 was caused by the Phe to Val mutation in the C-terminal of the signature motif, the Val to Phe mutant of HK1, namely HK1-V7F, was synthesized and tested for its bioactivity on SLα regulation in carp pituitary cells. In this case, the stimulatory effects of the HK1-V7F with the regeneration of the original “FXGLM” motif on SLα mRNA expression was markedly enhanced to the levels for HK2. These findings, as a whole, indicated that (i) the F residue in the “FXGLM” motif played a crucial role in the bioactivity of tachykinins in the fish model, and (ii) the Phe to Val mutation form in the “VXGLM” motif represented the molecular determinant in HK1 leading to weak agonistic actions on SLα regulation, presumably by interfering the interaction with target receptors. Our findings were also in agreement with the previous studies in mammalian cell models, e.g., COS-7 cells with NK3 receptor [9,12], confirming that the C-terminal “FXGLM” motif was essential for the bioactivity and receptor binding [32,50].

In rats, central administration of SP and EKA/B is known to induce pain-related behaviors, e.g., scratching paw withdrawal and thermal hyperalgesia [26,51], mainly through activation of NK1R expressed in the brain and/or spinal cord [52]. However, these stimulatory effects can be blocked by simultaneous treatment with EKC/EKD, and apparently, the “antagonistic effects” on NK1R activation induced by SP or EKA/B are mediated through the “FXGLL” motif of EKC/EKD [26,53,54]. These previous findings in mammalian models indicated that the Phe to Val mutation found in the “VXGLM” motif of HK1 might also have a functional impact on the receptor interaction and/or biological actions of other tachykinins in fish species. However, our present study found that HK1 could not block HK2-induced UTS1, PRL, and SLα mRNA expression. These results suggested that the V to F substitution in the signature motif of HK1 did not contribute to its “antagonistic effect” for HK2 on NKR binding/activation.

In summary, in order to clarify the pituitary actions of TAC4 gene products in lower vertebrates, the TAC4 and six NKR isoforms were isolated from grass carp. Sequence analysis showed that grass carp TAC4 could encode two mature peptides (namely HK1 and HK2), in which HK2 retained the typical FXGLM motif of tachyinin in the C-terminal, while HK1 contained a mutant VFGLM motif. The ligand-receptor selectivity showed that HK2 could activate all six NKR isoforms but with the highest activity for NK2R. Interestingly, HK1 could display a very weak activation for NK2R and NK3Ra2. Using grass carp pituitary cells as a model, we found that HK2 could significantly induce PRL, SLα, UTS1, NMB1, and CART mRNA expression mediated by NK2R and NK3R coupled with cAMP/PKA, PLC/IP3/PKC, and Ca^2+^/CaM/CaMK II cascades (Figure 10). These results indicated that HK2 might be involved in the regulation of reproduction, growth metabolism, and feeding in teleost. However, the potency and efficacy of HK1 was found to be ineffective for the six pituitary genes’ expression. Further studies indicated that the F to V mutation in “VXGLM” motif in HK1 contributed to its weak pituitary actions. Furthermore, we found that HK1 could not contribute to its “antagonistic effect” for HK2 on NKR binding/activation.

## Figures and Tables

**Figure 1 ijms-22-12893-f001:**
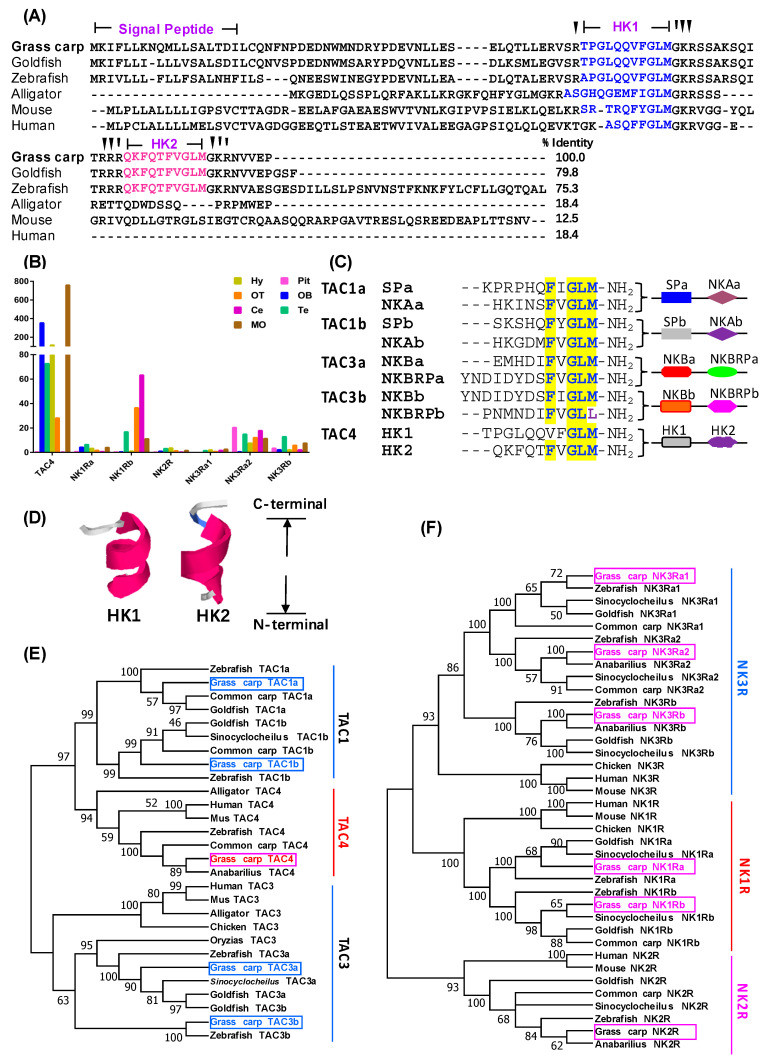
Sequence analysis of tachykinin 4 (TAC4) and six neurokinin receptors (NKRs) in grass carp. (**A**) Sequence analysis of TAC4 preprohormones among different vertebrates. The protein sequence of grass carp TAC4 was aligned with that reported in other vertebrates. The two mature peptides residues in these sequences were labeled in blue or red color, respectively. (**B**) Transcript level of TAC4 and six NKRs in various brain subregions and pituitary. Total RNA was extracted from various brain areas or pituitary of grass carp, and RT-PCR was performed by using specific primers for grass carp TAC4 and NKRs. (**C**) Sequences were compared among ten tachykinin peptides in grass carp. (**D**) Ribbon representation of grass carp HK1 and HK2 structural model. The amino acids with hydrophobic side chains were colored blue, while those with hydrophilic side chains were colored red. (**E**) The phylogenetic analysis of vertebrate TAC4 was performed using the neighbor-joining method (MEGA 6.0), and the grass carp TAC4 was highlighted in the red box. (**F**) The phylogenetic analysis of vertebrate six NKRs was performed by using the neighbor-joining method (MEGA 6.0).

**Figure 2 ijms-22-12893-f002:**
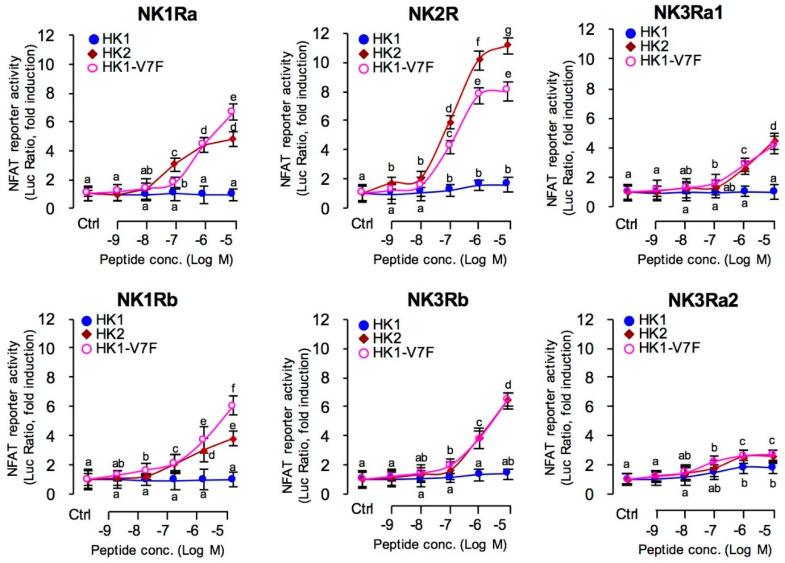
Receptor selectivity of HK1, HK2, and HK1-V7F. The nuclear factor of activated T-cells (NFAT)-Luc reporter together with NK1Ra, NK1Rb, NK2R, NK3Rb, NK3Ra1, or NK3Ra2, were transiently transfected into HEK-293 cells, respectively. The cells were incubated with increasing levels of grass carp HK1, HK2, or HK1-V7F (0.1–1000 nM), respectively. The luciferase activities were normalized against renilla expression in the same sample to adjust for potential variations in transfection efficiency. Each point is determined quadruplicate and is given as a mean ± SEM. The groups denoted by different letters represent significant differences at *p*-value < 0.05 (ANOVA followed Dunnett’s test).

**Figure 3 ijms-22-12893-f003:**
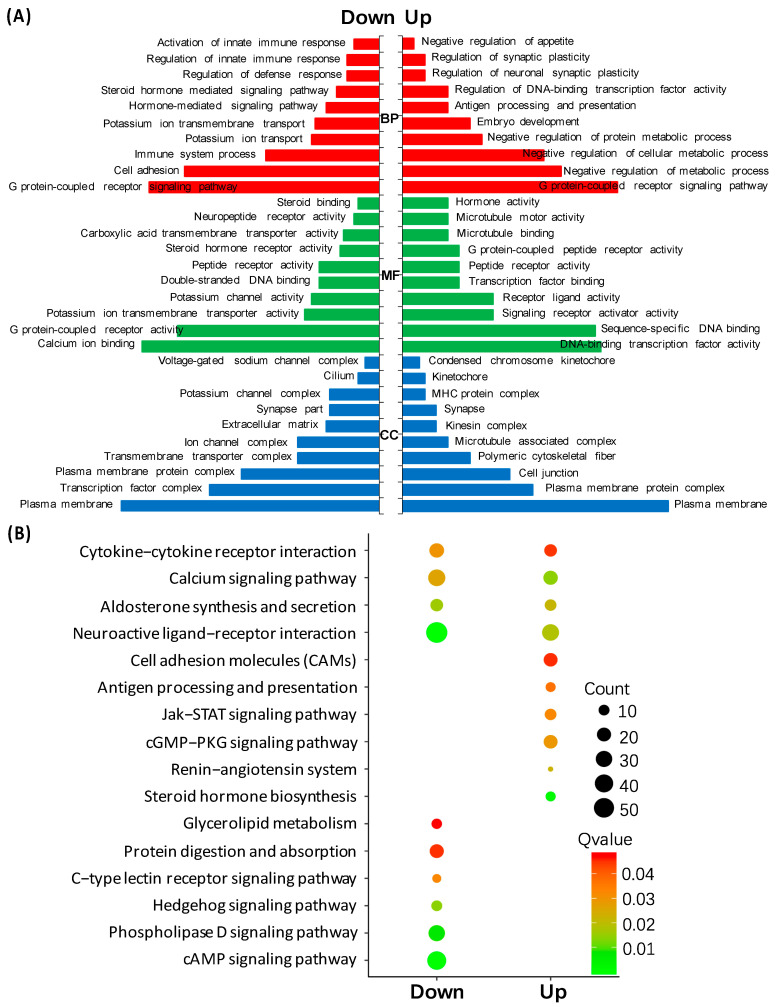
Gene ontology (GO) and Kyoto Encyclopedia of Genes and Genomes (KEGG) analysis of HK2-regulated different expressed genes (DEGs) in grass carp pituitary cells. (**A**) In GO analysis, DEGs were composed of three parts: cell component (CC), biological process (BP), and molecular function (MF). (**B**) KEGG analysis were used to enrich the top 10 KEGG pathways, including up-regulated and down-regulated pathways, respectively. Up, up-regulated genes; down, down-regulated genes; count, the number of DEGs.

**Figure 4 ijms-22-12893-f004:**
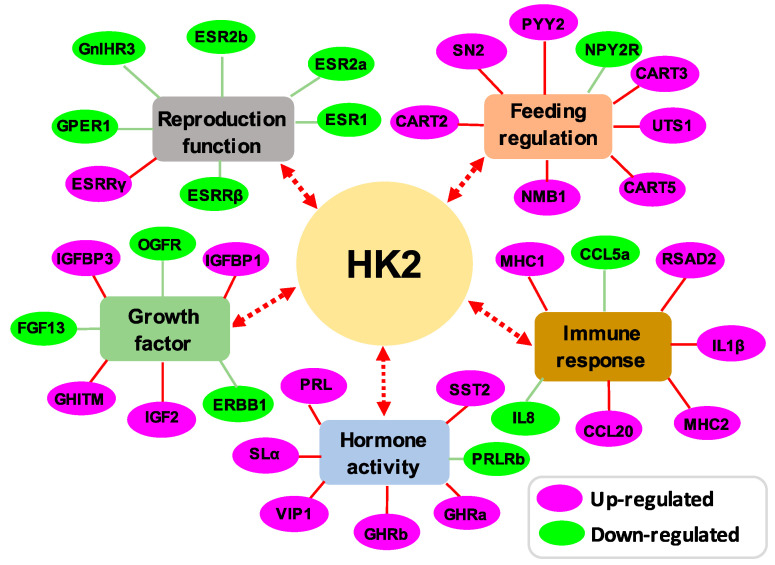
DEGs were enriched in the reproduction function, feeding regulation, growth factor, immune response, and hormone activity of trashcription. The pink color indicates upregulated DEGs, while the green color indicates downregulated DEGs.

**Figure 5 ijms-22-12893-f005:**
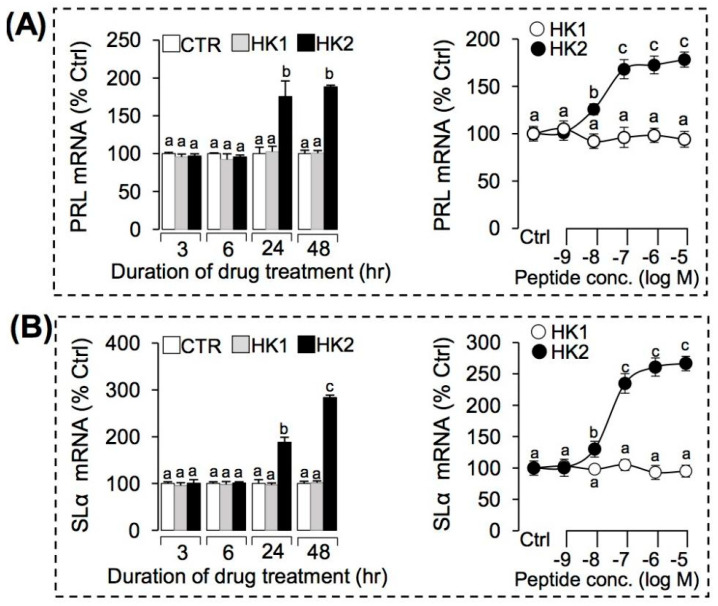
TAC4 gene products (HK1 and HK2) induced PRL (**A**) and SLα (**B**) mRNA expression in grass carp pituitary cells. Time course of grass carp HK1 (1 μM) and HK2 (1 μM) on PRL (**A**) and SLα (**B**) mRNA expression in grass carp pituitary cells. Dose-dependence of 24-h treatment with increasing levels of HK1 and HK2 (0.1–1000 nM) on PRL (**A**) and SLα (**B**) mRNA expression in grass carp pituitary cells. After drug treatment, the total RNA of the cells was extracted by the Trizol method, and the expression of various genes was detected by RT-PCR. Data presented were expressed as mean ± SEM, and the differences between groups were significant at *p*-value < 0.05 by labeling diverse letters.

**Figure 6 ijms-22-12893-f006:**
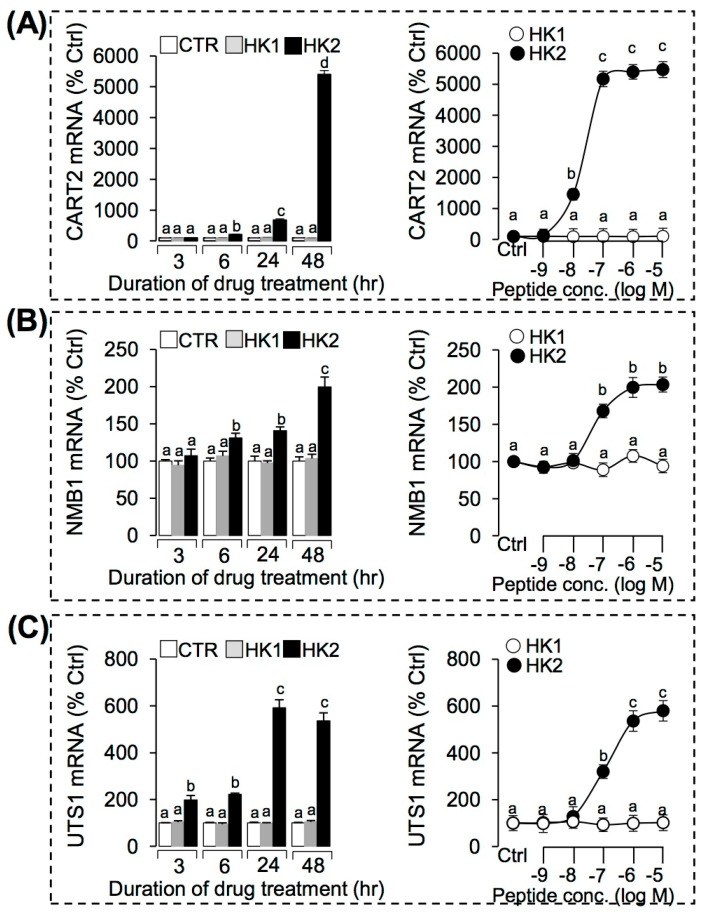
TAC4 gene products (HK1 and HK2) induced CART2, NMB1, and UTS1 mRNA expression in grass carp pituitary cells. Time course of grass carp HK1 (1 μM) and HK2 (1 μM) on CART2 (**A**), NMB1 (**B**), and UTS1 (**C**) mRNA expression in grass carp pituitary cells. Dose-dependence of 24-h treatment with increasing levels of HK1 or HK2 (0.1–1000 nM) on CART2 (**A**), NMB1 (**B**), and UTS1 (**C**) mRNA expression in grass carp pituitary cells. After drug treatment, the total RNA of the cells was extracted by the Trizol method, and the expression of various genes were detected by RT-PCR. Data presented were expressed as mean ± SEM, and the differences between groups were significant at *p*-value < 0.05 by labeling diverse letters.

**Figure 7 ijms-22-12893-f007:**
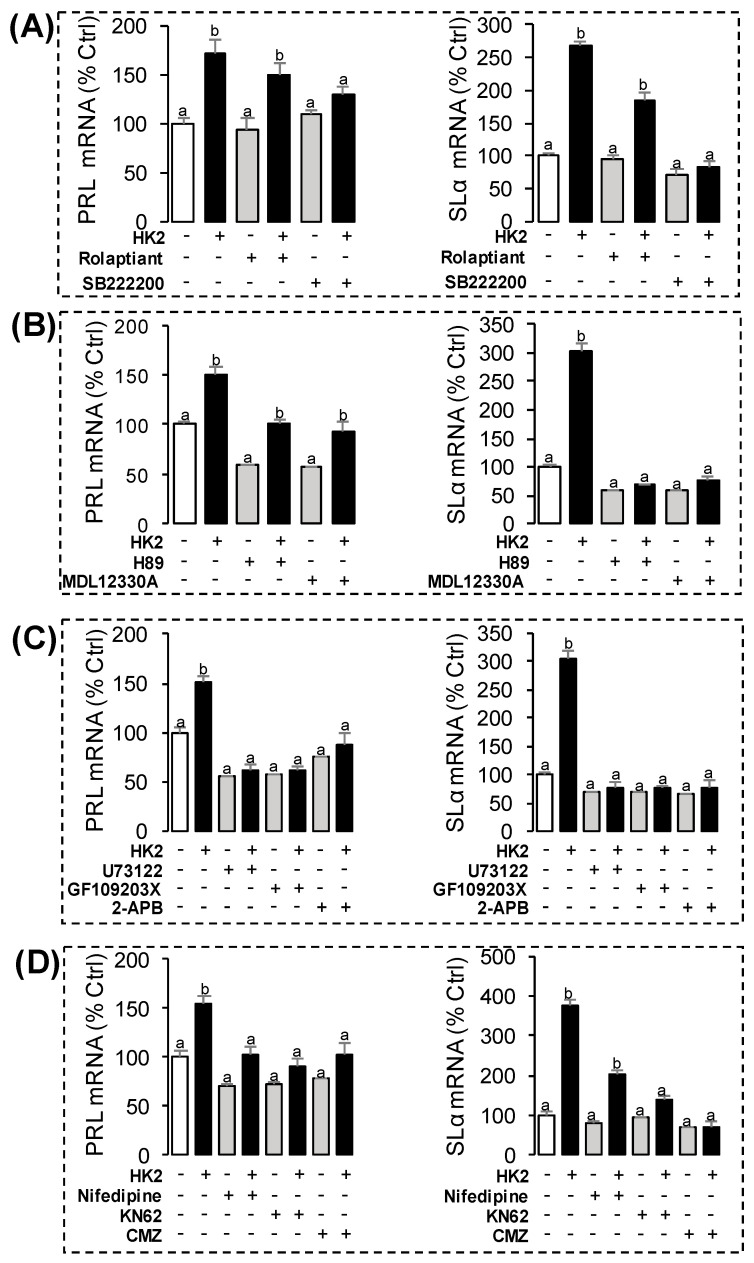
The receptor specificity and signal transduction for PRL and SLα regulation by HK2 in grass carp pituitary cells. (**A**) Effects of NKR subtype-specific antagonists on HK2-induced PRL and SLα mRNA expression. Pituitary cells were challenged for 24 h with grass carp HK2 (1 μM) in the presence or absence of the NK1R antagonist Rolaptiant (10 μM) or NK3R antagonist SB22200 (10 μM), respectively. (**B**) Effects of 24-h co-treatment with AC inhibitor MDL12330A (10 μM) or PKA inhibitor H89 (10 μM) on HK2-induced PRL and SLα mRNA expression. (**C**) Effects of 24-h co-treatment with PLC inhibitor U73122, PKC inhibitor GF109203X (10 μM), or IP_3_ receptor blocker 2-APB (10 μM) on HK2-induced PRL and SLα mRNA expression. (**D**) Effects of 24-h co-treatment with the VSCC inhibitor nifedipine (10 μM), CaM antagonist calmidazolium (1 μM) or CaMK-II blocker KN62 (10 μM) on HK2-induced PRL and SLα mRNA expression. After drug treatment, the total RNA of the cells was extracted by the Trizol method, and the expression of various genes was detected by RT-PCR. Data presented were expressed as mean ± SEM, and the differences between groups were significant at *p*-value < 0.05 by labeling diverse letters.

**Figure 8 ijms-22-12893-f008:**
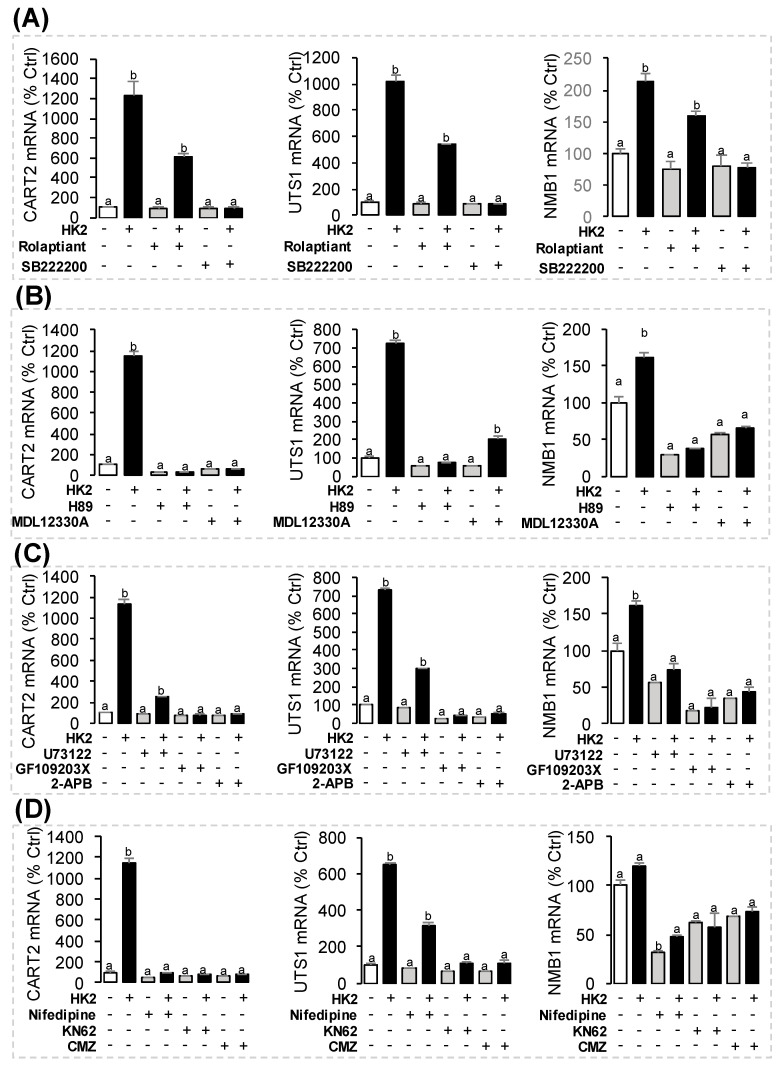
The receptor specificity and signal transduction for CART2, NMB1, and UTS1 regulation by HK2 in grass carp pituitary cells. (**A**) The pituitary cells were treated for 24 h with grass carp HK2 (1 μM) in the presence or absence of NK1R antagonist Rolaptiant (10 μM) or NK3R antagonist SB222200 (10 μM). (**B**) Effects of 24-h co-treatment with an AC inhibitor MDL12330A (10 μM) or PKA inhibitor H89 (10 μM) on HK2-induced CART2, UTS1, or NMB1 mRNA expression. (**C**) Effects of 24h co-treatment with a PLC inhibitor U73122, PKC inhibitor GF109203X (10 μM), or IP_3_ receptor blocker 2-APB (10 μM) on HK2-induced CART2, UTS1 or NMB1 mRNA expression. (**D**) Effects of 24-h co-treatment with the VSCC inhibitor nifedipine (10 μM), CaM antagonist calmidazolium (1 μM), or CaMK-II blocker KN62 (10 μM) on HK2-induced CART2, UTS1 or NMB1 mRNA expression.

**Figure 9 ijms-22-12893-f009:**
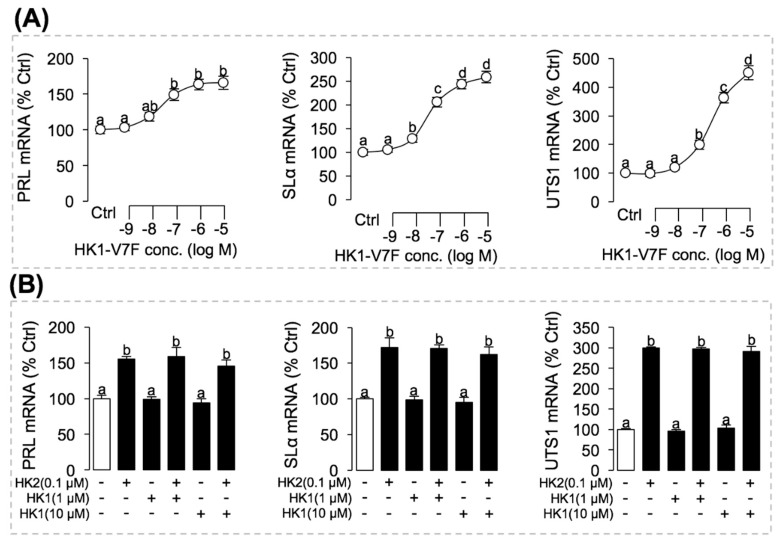
Phe to Val mutation in FXGLM motif in PRL, SLα, and UTS1 regulation by HK2 in grass carp pituitary cells. (**A**) Effects of HK1-V7F on PRL, SLα, and UTS1 mRNA expression in grass carp pituitary cells. In this experiment, grass carp pituitary cells were incubated for 24 h with increasing levels of HK1-V7F (0.1–1000 nM). (**B**) Suppressed effects of HK1 on HK2-induced PRL, SLα, and UTS1 mRNA expression in grass carp pituitary cells. Pituitary cells were challenged for 24 h with grass carp HK2 (0.1 µM) in the presence or absence of HK1 (1 µM) and HK1 (10 µM). In these experiment, Trizol method was used to extract total cell RNA, and RT-PCR was used to detect PRL, SLα, and UTS1 mRNA expression. Data presented are expressed as mean ± SEM, and the differences between groups were significant at *p*-value < 0.05 by labeling diverse letters.

**Figure 10 ijms-22-12893-f010:**
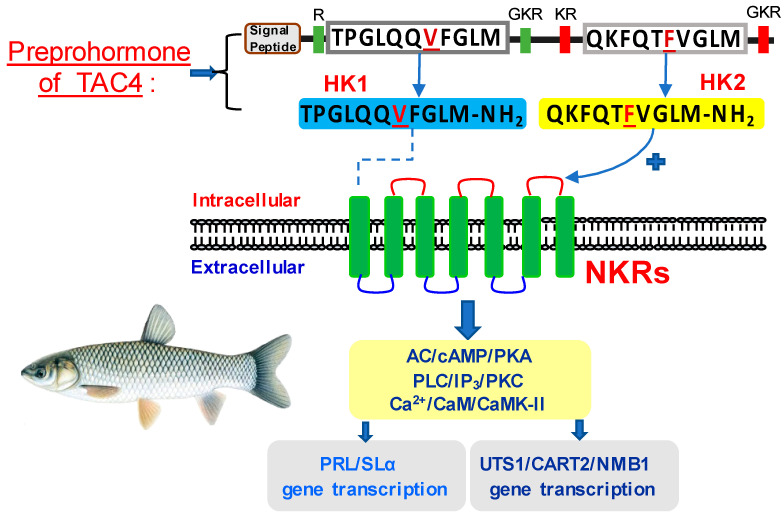
Working model of TAC4 gene products in the pituitary. TAC4 could encode two mature peptides, namely HK1 and HK2. HK2 could stimulate PRL, SLα, UTS1, NMB1, and CART2 mRNA expression mediated by NK2R and NK3R coupled with cAMP/PKA, PLC/IP3/PKC, and Ca^2+^/CaM/CaMK II cascades. In contrast, HK1 was found to be ineffective for the six pituitary genes’ expression.

**Table 1 ijms-22-12893-t001:** The up-regulated genes induced by HK2 in grass carp pituitary cells.

Gene	Description	FC	Go-Biological Process
CART2	Cocaine and amphetamine-regulated transcript 2 precursors	27.23	Negative regulation of appetite
CART5	Cocaine and amphetamine-regulated transcript 5 precursors	1.23	Negative regulation of appetite
CKB	Brain creatine kinase b	1.63	Creatine Kinase Activity
CRFB2	Cytokine receptor family member b2 precursor	1.77	Interferon receptor activity
GHITM	Growth hormone-induced transmembrane protein	1.21	The components of the membrane
GPR186	G protein-coupled receptor 186	1.83	Signal sensor activity
IGFBP1	Insulin-like growth factor-binding protein 1	1.59	Cell growth regulation
MPR63	Mitochondrial ribosomal protein 63	1.35	Mitochondrial Ribosome
NLRP12	Proteins containing NACHT, LRR, and PYD domains 12	2.73	ATP binding
NMB1	Neurotonin B1	1.50	Neuropeptide signaling pathway
NPAS4	Neuron contains PAS domain protein 4	2.69	DNA binding
NSFb	N-ethylmaleimide sensitive factor b	1.94	ATP binding
PRL1	Prolactin 1	1.71	Hormonal activity
RBM24	RNA binding protein 24	2.92	Nucleotide binding
SDC2	Heparan sulfate proteoglycan precursor	1.62	Gastrointestinal development
SERPINE2	Glial cell-derived neuron precursor	1.33	Extracellular space
SID4	Secreted immunoglobulin domain 4 precursor	3.89	Same protein binding
SLα	Somatostatin alpha subunit	1.81	Hormonal activity
SN2	Secretagogue 2 precursor	1.78	Secretory granules
UTS1	Urotensin 1	7.31	Hormonal activity

FC: fold change; FDR: false discovery rate.

**Table 2 ijms-22-12893-t002:** The down-regulated genes induced by HK2 in grass carp pituitary cells.

Gene	Description	FC	Go-Biological Process
AIG1	Androgen inducible protein 1	0.77	The components of the membrane
CALCRLb	Calcitonin gene-related peptide-like receptor b	0.75	Calcitonin receptor activity
DHRS13	Dehydrogenase/reductase SDR family member 13	0.43	Oxidoreductase activity
FGF14	Fibroblast Growth Factor 14	0.80	Growth factor activity
FGFR2	Fibroblast Growth Factor Receptor 2	0.46	Fibroblast growth factor binding
GHSR1	Growth hormone secretagogue receptor type 1	0.14	Signal sensor activity
GnIHR3	Gonadotropin inhibits hormone receptor 3	0.68	Receptor activity
GPER1	G protein-coupled estrogen receptor 1	0.81	G protein-coupled receptor activity
GRB2a	Growth factor receptor-binding protein 2a	0.77	Binding activity
IGF1Ra	Insulin-like growth factor type 1 receptor a	0.68	Insulin-like growth factor binding
INO1	Inositol-3-phosphate synthase 1	0.78	Inositol-3-phosphate synthase activity
MESD	LDLR molecular chaperone MESD	0.69	Wnt signaling pathway
NRP2b	Neurocin 2b	0.71	Metal ion binding
OGFR	opioid growth factor receptor-like protein 1	0.36	Receptor activity
OLFM2	Olfactory protein 2	0.66	Neural c cell development
S100A10	S100 Calbindin A10b	0.60	Calcium ion binding
SAA	Serum amyloid A precursor	0.72	Response to bacteria
SSTR2a	Somatostatin receptor 2	0.54	Somatostatin receptor activity
SSTR3	Somatostatin Receptor Type 3	0.57	Somatostatin receptor activity
ZMYND11	Zinc finger MYND domain-containing protein 11	0.49	Zinc ion binding

FC: fold change; FDR: false discovery rate.

## Data Availability

The original contributions presented in the study are included in the article/Appendix A. Further inquiries can be directed to the corresponding author.

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
