# Peer review of "Novel Pituitary Actions of TAC4 Gene Products in Teleost"

_ijms, 2021, doi:10.3390/ijms222312893_

Round 1

Reviewer 1 Report

In this manuscript, Shi et al. present a series of different studies revolving around the grass carp TAC4 gene and its products, notably HK1 and HK2. Quintessentially they show that HK1 in teleost harbors a mutation that renders it basically inoperable (so to speak). Both peptides and the potentially targeting neurokinin receptors are thoroughly analyzed in order to decipher which combinations lead to downstream signalling and which pathways are involved. The results are then comprehensibly discussed and even though the clear physiological implications are to be elucidated, some hypotheses based on prior work from other research groups are formulated. The senior authors that serve as corresponding authors have a long-standing experience in investigating teleost and pituitary hormones (cf. http://sourcedb.ihb.cas.cn/cn/expert/200907/t20090722_2156194.html, https://zfin.org/ZDB-LAB-110330-2, and https://www.researchgate.net/profile/Guangfu-Hu-2). Many studies from their labs have been previously and recently published in diverse journals and an overarching theme is recognizable.

The work is clearly presented and the manuscript is mostly easy to read and follow. A few phrases are somewhat bumpy and reveal the non-native English speaking nature of the authors. Throughout the manuscript there are numerous spelling mistakes as well. It might be a good idea to have a knowledgeable English native-speaker proofread the manuscript.

The introduction is well founded on existing knowledge, but I found it to jump back and forth between topics. I would suggest to cover the aspects involving TAC3 first (if at all; maybe this part could be omitted?) and then TAC4.

The methods are clearly described and all steps are described with enough detail to allow for the study to be reproduced.

One thing that I found difficult to grasp in the results section was the nomenclature involving HK1 and its mutation. Given the fact that the FXGLM motif is the "standard" amino acid sequence of tachykinins, the VXGLM variant in carp's HK1 is "exceptional" and in its own right a [well conserved] mutation. So speaking of HK1 (the mutated one) and HK1-M (the mutation mutated back to the usual wildtype = the "normal" one) is somehow confusing. Maybe it would be better to choose other terms to describe those so that it becomes less confusing to read the results. Furthermore, the introductory sentences from section 3.7 should be move to the start of 3.1.

Overall, the work is very well presented and -as far as can be told- meticulously performed. The manuscript deserves to be published in the International Journal of Molecular Sciences, but there are some minor revisions that need to be performed first.

Revisions to be made:
* restructure introduction
* name HK1 and HK1-M differently, make this more clear in the results
* spell out ALL (!) abbreviations at first use - don't leave any out
* proof-reading by English native-speaker (please cf. the attached scan of handwritten annotations for some pointers)

Author Response

Comments and Suggestions for Authors

Overall, the work is very well presented and as far as can be told- meticulously performed. The manuscript deserves to be published in the International Journal of Molecular Sciences, but there are some minor revisions that need to be performed first.

Reponses:

Thank you very much for your help with the paper review along the way. As you suggested, we have revised the manuscript again. Thanks a lot again.

Q1. restructure introduction

Response:

Thank you very much for your suggestion. As you suggested, we restructure the Introduction, and introduce tachykinin family following as TAC1, TAC3 and TAC4.  

“In mammals, the tachykinin (TAC) family includes three members, namely TAC1, TAC3 (TAC2 in rodents) and TAC4, respectively. Among them, TAC1 gene encodes substance P (SP) and neurokinin A (NKA). SP is known to be involved in the regulation of pain control/injury [1], neurogenicin-flammation [2] and obesity [3] with mammal. Compared with mammals, teleost have undergone an additional genome duplication during evolution, which is called fish specific genome duplication (FSGD) or the 3R hypothesis [4]. So there were two isoforms TAC1 (namely TAC1a and TAC1b) and TAC3 (namely TAC3a and TAC3b), respectively. Similar to mammal, teleost TAC1 also encoded two mature peptides, namely SP and NKA, respectively [5]. In teleost, SP and NKA can trigger the secretion of luteinizing hormone (LH), prolactin (PRL) and somatostatinα (SLα) in the pituitary cells of carp, and the transcriptional levels of PRL and SLα increase in parallel. Short-term SP treatment (3h) induced the release of LH, while prolonged induction time (24h) inhibited LHβ mRNA expression [5].

In mammals, TAC3 encodes neurokinin B (NKB), which is known to be involved in the regulation of smooth muscles of the gastrointestinal tract, secretion of intestinal epithelial fluid, vasodilation, and stimulating sperm motility [6-8]. Recent studies have found that NKB has become a key regulator of mammalian reproductive function, especially controlling the release of gonadotropin-releasing hormone (GnRH) in the hypothalamus [9,10]. In contrast to mammals, TAC3 did not only encode NKB, but also a novel tachykinin peptide in teleost, named neurokinin B-related peptide (NKBRP) [11]. The research on the function of TAC3 gene products in fish mainly focuses on the regulation of reproductive process. Previous studies have found that intraperitoneal injection of NKB or NKBRP could both induce LHβ secretion in sexually mature female zebrafish [12]. In addition, our previous studies have shown that both NKB and NKBRP could promote PRL and SLα secretion and mRNA synthesis in grass carp pituitary cells [11].

Compared to TAC1 and TAC3, TAC4 is the last member of the tachykinin family. In 2000, Zhang et al. was firstly isolated the TAC4 gene from mouse hematopoietic stem cells , and named its gene production as hemokin 1 (HK1) [13]. Subsequently, TAC4 were cloned from rats and humans. Among them, the mouse hemokinin 1 (mHK1) encoded by the rat TAC4 gene has extremely high similarity with rat hemokinin 1 (rHK1), while the human race TAC4 gene encoding product was fairly different [14]. The TAC4 gene can encode six neuropeptides, namely HK1, HK1 (4-11), endokinin A (EKA), endokinin B (EKB), endokinin C (EKC) and endokinin D (EKD), respectively [15-17]. In mammals, recent studies have shown that HK1 and EKs were new mediators of pain and inflammation, and also play the crucial role in the hematopoietic system, anti-anxiety and anti-depression [18-20]. However, little information is available about TAC4 in teleost.

In the present study, using grass carp (Ctenopharyngodon idellus) as model, we try to examine the pituitary actions of TAC4 gene product in teleost. Firstly, the grass carp TAC4 were cloned, which encoded two mature peptides, namely HK-1 and HK-2, respectively. Secondly, six potential NKRs were isolated from grass carp. Then, by using transfection and dual-luciferase detection, we tried to confirm the specific receptor for HK-1 and HK-2, respectively. Thirdly, using grass carp pituitary cells as model, we try to examine the direct pituitary actions of HK-1 and HK-2 in teleost. Finally, we try to clarify the mechanism of functional differences between HK-1 and HK-2 in teleost.   

  1. Lin, C.; Chen, W.N.;Chen, C.J.; Lin, Y.W.; Zimmer, A.; Chen, C.C. An antinoci-ceptive role for substance P in acid-induced chronic muscle pain. J. Pro. Nat. Acad. Sci. Unit. States. Ame. 2012, 109, E76-83.
  2. Ang, S.F.; Moochhala, S.M.; Macary, P.A.; Bhatia, M. Hydrogen sulfide and neurogenic inflammation in polymicrobial sepsis: involvement of substance P and ERK-NF-κB signaling. J. PLoS. ONE. 2017, 6, e24535.
  3. Chitrang, T.; Shan, X.; Loraine, Y.C.; Dhiraj, K.; Jenna, H.; Sarah A.; Kristy, H.;Henriette, K.; Yeo, G.S.H.; Diego, P.T. Tachykinin-1 in the central nervous system regulates adiposity in rodents. J. Endocrinology. 2015, 156, 1714-23.
  4. Meyer, A.; Schartl, M. Gene and genome duplications in vertebrates:the one-to-f-our (-to-eight in fish) rule and the evolution of novel gene functions. J. Curr. Opin. Cell. Bio. 1999, 11, 699-704.
  5. Hu, G.F.; He, M.; Ko W.; Wong A. TAC1 GeneProducts RegulatePituitary Hormone Secretion and Gene Expression in Prepubertal Grass Carp Pituitary Cells. J. Endocrinology. 2017, 158, 1776-1797.
  6. Abdelrahman, A.M.; Pang, C. Effect of Substance P on Venous Tone in Conscious Rats. J. Jou. Cardio. Pharma. 2005, 45, 49.
  7. Lecci, A.; Capriati, A.; Altamura, M.; Maggi, C.A. Tachykinins and tachykinin receptors in the gut, with special reference to NK2 receptors in human. J. Auto. Neur. 2006, 126, 232-249.
  8. Bae, Y.K.; Kani, S.; Shimizu, T.; Tanabe, K.; Nojima, H.; Kimura, Y.; Higashijima, S.I.; Hibi, M. Anatomy of zebrafish cerebellum and screen for mutations affecting its development. J. Deve. Bio. 2009, 330, 406-426.
  9. Topaloglu, A.K.; Reimann, F.; Guclu, M.; Yalin, A.S.; Semple, R.K. Tac3 and tacr3 mutations in familial hypogonadotropic hypogonadism reveal a key role for neurokinin b in the central control of reproduction. J. Nat. Gene. 2009, 41, 354-358.
  10. Goodman, R.L; Coolen, L.M.; Lehman, M.N. A role for neurokinin b in pulsatile gnrh secretion in the ewe. J. Neuroe. 2014, 99, 18.
  11. Hu, G.F.; Lin, C.; He, M.; Wong, A. Neurokinin B and reproductive functions: “KNDy neuron” model in mammals and the emerging story in fish. J. Gene. Comp. Endocrinology. 2014, 208, 94-108.
  12. Biran, J.; Palevitch, O.; Ben-Dor, S.; Levavi-Sivan, B. Neurokinin Bs and neurokinin B receptors in zebrafish-potential role in controlling fish reproduction. J. Proce. Nat. Acad. Sci. Unit. States. Ame. 2012, 109, 10269-74.
  13. Zhang, Y.; Lu, L.; Furlonger, C.; Wu, G.; Paige, C. Hemokinin is a hematopoieticspecific tachykinin that regulates B lymphopoiesis. J. Nat. Immunology. 2000, 1,392-397.
  14. Kurtz, M.M.; Wang, R.; Clements, M.K.; Cascieri, M.A.; Austin, C.P.; Cunningh-am, B.R.; Chicchi, GG.; Liu, Q. Identification, localization and receptor characterization of novel mammalian substance P-like peptides. J. Gen. 2002, 296, 205-212.
  15. Page, N.M.; Bell, N.J.; Gardiner, S.M.; Manyonda, I.T.; Brayley, K.J.; Strange, P.G.; Lowry, PJ. Characterization of the endokinins: human tachykinins with cardiovascular activity. J. Pro. Nat. Acad. Sci. Unit. States. Ame. 2003, 100, 6245-50.
  16. Steinhoff, M.S.; Mentzer, B.V.; Geppetti, P.; Pothoulakis, C.; Bunnett, N.W. Tac-hykinins and their receptors: contributions to physiological control and the mechanisms of disease. J. Phy. Rev. 2014, 94, 265-301.
  17. Zieglgänsberger, W. Substance P and pain chronicity. J. Cell. Tiss. Res. 2019, 375, 227-241.
  18. Dai, L.; Perera, D.S.; King, D.W.; Southwell, B.R.; Burcher, E.; Liu, L.; Hemoki-nin-1 stimulates prostaglandin E₂ production in human colon through activationof cyclooxygenase-2 and inhibition of 15-hydroxyprostaglandin dehydrogenase. J.Phar. Expe. Therapeutics. 2012, 340, 27-36.
  19. Tsilioni, I.; Russell, J.; Stewart, R.; Gleason, T, Theoharides, T. Theoharides Neuropeptides CRH, SP, HK-1, and Inflammatory Cytokines IL-6 and TNF Are Increased in Serum of Patients with Fibromyalgia Syndrome, Implicating Mast Cells. J. Pha. Expe. Therapeutics. 2016, 356, 664-72.
  20. Borbély, É.; Helyes, Z. Role of hemokinin-1 in health and disease. J. Neuropept-des. 2017, 64, 9-17.

Q2. name HK1 and HK1-M differently, make this more clear in the results

Response:

Thank you very much for your suggestion. We kept the naming of HK1 unchanged, and named HK1-M as HK1-V7F. HK1-V7F means that the seventh amino acid valine (V) of the mature peptide HK1 is changed to phenylalanine (F), which restores the classic sequence FXGLM of the tachykinin family.

Q3. spell out ALL (!) abbreviations at first use - don't leave any out
Response:

Thank you very much for your suggestion.

We have revised “TAC4” to “Tachykinin 4 (TAC4)”. (Line 10)

We have revised “HK1” to “Hemokinin 1(HK1)”. (Line 14)

We have revised “HK2” to “Hemokinin 2(HK2)”. (Line 14)

We have revised “NK2R” to “Neurokinin receptor 2 (NK2R)”. (Line 17)

We have revised “PRL” to “Prolactin (PRL)”. (Line 19)

We have revised “SLα” to “Somatolactin α (SLα)”. (Line 19)

We have revised “UTS1” to “Urotensin 1(UTS1)”. (Line 19)

We have revised “NMB1” to “Neuromedin-B 1 (NMB1)”. (Line 19)

We have revised “CART2” to “Cocaine- and amphetamine-regulated transcript (CART2)”. (Line 20)

We have revised “NK3R” to “Neurokinin receptor 3 (NK3R)”. (Line 21)

We have revised “cAMP” to “Cyclic adenosine monophosphate (cAMP)”. (Line 21)

We have revised “PKA” to “Protein kinase A (PKA)”. (Line 21)

We have revised “PLC” to “Phospholipase C (PLC)”. (Line 22)

We have revised “IP3” to “Inositol 1,4,5-triphosphate (IP3)”. (Line 22)

We have revised “PKC” to “Protein kinase C (PKC)”. (Line 22)

We have revised Ca2+” to “calcium2+ (Ca2+)”. (Line 22)

We have revised “CaM” to “Calmodulin (CaM)”. (Line 23)

We have revised “CaMK II” to “Calmodulin kinase-II (CaMK II)”. (Line 23)

We have revised “V” to “Valine (V)”. (Line 25)

We have revised “F” to “Phenylalanine (F)”. (Line 25)

We have revised RT-qPCR” to “Real-time quantitative PCR (RT-qPCR)”. (Line 98)

We have revised “NK1Ra” to “Neurokinin receptor 1 isoforms a (NK1Ra)”. (Line 108)

We have revised “NK1Rb” to “Neurokinin receptor 1 isoforms b (NK1Rb)”. (Line 108)

We have revised “NK3Ra1” to “Neurokinin receptor 3 isoforms a1 (NK3Ra1)”. (Line 109)

We have revised “NK3Ra2” to “Neurokinin receptor 3 isoforms a2 (NK3Ra2)”. (Line 109)

We have revised “NK3Rb” to “Neurokinin receptor 3 isoforms b (NK3Rb)”. (Line 110)

We have revised “GFP” to “Green fluorescent protein (GFP)”. (Line 113)

We have revised “SSTR2a” to “Somatostatin receptor Type 2 isoforms a (SSTR2a)”. (Line 241)

We have revised “SSTR3” to “Somatostatin Receptor Type 3 (SSTR3)”. (Line 241)

We have revised “OGFR” to “Opioid growth factor receptor-like protein 1 (OGFR)”. (Line 242)

We have revised “GHSR1” to “Growth hormone secretagogue receptor type 1 (GHSR1)”. (Line 243)

We have revised “FGFR2” to “Fibroblast Growth Factor Receptor 2 (FGFR2)”. (Line 243)

We have revised “GH” to “Growth hormone (GH)”. (Line 435)

Q4. proof-reading by English native-speaker (please cf. the attached scan of handwritten annotations for some pointers)
Response:

Thank you very much for your suggestion. The amendments we made are marked in yellow in the submitted manuscript, and the amendments made are listed below

We have revised “The ligand-receptor selectivity showed that HK2 could activate all six NKR isoforms but with the highest activity for the NK2R.” to “The ligand-receptor selectivity showed that HK2 could activate all 6 NKRs but with the highest activity for the neurokinin receptor 2 (NK2R).”. (Line 16)

We have revised “In grass carp pituitary cells, HK2 treatment was shown to induce PRL,” to “In grass carp pituitary cells, HK2 could induce prolactin (PRL),”. (Line 18)

We have revised “However, the corresponding stimulatory effects triggered by HK1 treatment were found to be notably weaker.” to “However, the corresponding stimulatory effects triggered by HK1 were found to be notably weaker.”. (Line 23)

We have revised “Two-year-old grass carp with a body weight of 1.7±0.2 kg was purchased from local market and kept in a well-aerated 250-liter aquaria under a 12-hour light/12-hour dark photoperiod. And then kept in the aquaria at 28±1 for seven days and without feeding for at least the 3 days.” to “Two-year-old grass carps with a body weight of 1.7±0.2 kg were purchased from local market and kept in a well-aerated 250-liter aquaria under a 12-hour light/12-hour dark photoperiod at 28±1 for seven days.”. (Line 84)

We have revised “In view of the grass carp at this stage was prepuberal and the sex character was not obvious, so breed fish with mixed sexes for pituitary cell preparation.” to “Grass carps at this stage were prepuberal and the sex character was not obvious, so breed fish with mixed sexes for pituitary cell preparation.”. (Line 86)

We have revised “Grass carp HK1, HK2 and HK1-M, were synthesized by GeneScript Corporation (Piscataway, NJ). HK1 (TPGLQQVFGLM-NH2), HK2 (QKFQTFVGLM-NH2) and HK1-M (TPGLQQFFGLM-NH2) short peptides were dissolved in ultrapure water and divided into 1 mM stocks, and stored at -80.”to “Grass carp HK2 (QKFQTFVGLM-NH2), HK1 (TPGLQQVFGLM-NH2), and HK1-V7F (TPGLQQFFGLM-NH2) were synthesized by GeneScript Corporation (Piscataway, NJ). The three mature peptides were subpackaged at 1 mM concentration using ultrapure water and stored at -80 .”. (Line 91)

We have revised “I In order to clarify the distribution of TAC4 and its receptors in different tissues of grass carp, total RNA were extracted from each brain subregion, pituitary, gland, and peripheral tissues of grass carp.” to “In order to clarify the distribution of TAC4 and its receptors in various tissues of grass carp, total RNA were extracted from each brain subregion, pituitary gland, and peripheral tissues of grass carp.”. (Line 96)

We have revised “After the concentration was detected by the Nanodrop 2000, each tissue took an equal amount of total RNA for reverse transcription to prepare a cDNA template.” to “Then the concentration was detected by the Nanodrop 2000, and each tissue took an equal amount of total RNA for reverse transcription to prepare a cDNA template.”. (Line 98)

We have revised In these studies, RT-PCR for β-actin was performed as an internal control. Sequence alignment and phylogenetic analysis of carp TAC4 and NKRs was conducted with ClustalX 2.1 and MEGA 7.0 . Based on the amino acid sequence deduced, protein structures for grass carp TAC4 and NKRs was predicted using I-TASSER and SWISS-MODEL ” to “In these studies, β-actin was performed as an internal control. Sequence alignment and phylogenetic analysis of TAC4 and NKRs was conducted with ClustalX 2.1 and MEGA 7.0 (http://www.megasoftware.net/). The 3D structures for TAC4 and NKRs were predicted by I-TASSE (https://zhanglab.ccmb.med.umich.edu/I-TASSER/) and SWISS-MODEL (https://swissmodel.expasy.org/), respectively.”. (Line 98)

We have revised “To clarify the receptor selectivity of TAC4 gene products, this study conducted transfection experiments in HEK293T cells, and detected the fluorescent signal by the firefly luciferase reporter gene detection kit.” to “To clarify the receptor selectivity of TAC4 gene products, transfection experiment was conducted in HEK293T cells, and the fluorescent signal was detected by the firefly luciferase reporter gene detection kit.”. (Line 105)

We have revised Then transfected HEK293T cells were incubated various concentration HK-1 (1-10000nM), HK-2 (1-10000nM), or HK1-M (1-10000nM) for 24 hours, respectively. Finnaly, the cell culture medium was removed, the cells were collected to detect the luciferace value according to the instructions of the firefly luciferase reporter gene detection kit.” to “The transfected HEK293T cells were incubated with various concentration HK-1 (1-10000nM), HK-2 (1-10000nM), or HK1-V7F (1-10000nM) for 24 hours, respectively. Finally, the transfected cells were collected to detect the luciferase value according to the instructions of the firefly luciferase reporter gene detection kit.

”. (Line 114)

We have revised “Grass carp pituitary cells were obtained by trypsin digestion [16]. Then, the pituitary cells were seeded into a 24-well cell culture plate at a density of 2.5×106 cells/well at 28°C under 5% CO2. When the cell connection was reached more than 90%, HK2 (1μm) was used to incubate the cells.” to “Grass carp pituitary cells were obtained by trypsin digestion [21]. Then, the pituitary cells were seeded into a 24-well cell culture plate at a density of 2.5×106 cells/welland incubated at 28°C under 5% CO2 in cell culture incubator. The primary culture pitutiary cells were treated by HK2 (1 μM) for 24 hours.”. (Line 121)

We have revised “Based on the negative binomial distribution model, the edgeR software was used to calculate the differential expression based on the FPKMFragments per kilobase of exon per million fragments mappedvalue of the gene.” to “Using the negative binomial distribution model, the edgeR software was applied to calculate the differential expression based on the FPKMFragments per kilobase of exon per million fragments mappedvalue of the gene.”. (Line 126)

We have revised “With the purpose of clarify the regulatory effect of HKs on the gene expression and protein secretion in grass carp pituitary, trypsin digestion method was used to prepare for grass carp primary cultured pituitary cells.” to “To further confirm the pituitary actions of HKs in grass carp, trypsin digestion method was used to prepare for grass carp primary cultured pituitary cells..”. (Line 135)

We have revised “After drug treatment, total RNA was extracted from adherent cells by Trizol, and then reverse transcription was performed by HifairTM III 1st Strand cDNA Synthesis Kit (Yeasen, Shanghai, China).” to “After drug treatment, total RNA was extracted from adherent cells by Trizol, and then subjected to reverse transcription using HifairTM III 1st Strand cDNA Synthesis Kit (Yeasen, Shanghai, China).”. (Line 140)

We have revised “After drug treatment, total RNA was extracted from adherent cells by Trizol, and then reverse transcription was performed by HifairTM III 1st Strand cDNA Synthesis Kit (Yeasen, Shanghai, China).” to “After drug treatment, total RNA was extracted from adherent cells by Trizol, and then subjected to reverse transcription using HifairTM III 1st Strand cDNA Synthesis Kit (Yeasen, Shanghai, China).”. (Line 140)

We have revised “In this studiesour using RT-qPCR to detect mRNA for TAC4, NKRs, PRL, SLα, CART2, UTS1, SN2 and NMB1.” to “In this study, RT-qPCR was used to detect mRNA expression for TAC4, NKRs, PRL, SLα, CART2, UTS1, SN2 and NMB1.”. (Line 148)

We have revised All RT-qPCR experiments use β-actin as the internal reference gene, and β-actin mRNA has no significant changes, the data of PRL, SLα, CART2, UTS1, SN2 and NMB1 are converted into average percentages in the control group (%Ctrl). In the transfection experiment, the detected firefly luciferase activity is routinely normalized with the GFP activity expressed in the same well, expressed as the luciferin activity ratio Luc Ratio. The data shown in the Figureureure is mean ± SEM, and the test data are from four parallel experiments. All the data are analyzed by ANOVA, and then Dunnett's test is performed with GraphPad Prism 7.0. P<0.05 indicates significant differences between groups.” to “Since the transcript level of β-actin showed no significant changes, the data of PRL, SLα, CART2, UTS1, and NMB1 were converted into average percentages in the control group (% Ctrl). In the transfection experiment, the detected firefly luciferase activity was routinely normalized with the GFP activity expressed in the same well, expressed as the luciferin activity ratio Luc Ratio. The data from four parallel experiments was presented as mean ± SEM in the figure. All the data were analyzed by ANOVA, and then subjected to Dunnett's test in GraphPad Prism 7.0. The significant differences between groups were indicated by P<0.05..”. (Line 151)

We have revised “Similar to other fish species, grass carp TAC4 also encoded two mature peptides, a 11-a.a HK2 carrying the tachykinin signature motif (FXGLM) and a 10-a.a. HK1 containing a mutated form form of tacykinin consensus motif form (VFGLM)” to “Sequence analysis showed that grass carp TAC4 encoded two mature peptides, including a 11-a.a HK2 carrying the classical tachykinin signature motif (FXGLM) and a 10-a.a HK1 containing a mutated form of tacykinin consensus motif form (VFGLM) (Figure 1A&C)”. (Line 163)

We have revised Finanlly, tissue distribution showed that TAC4 was widely distributed in various brain regions except cerebellum, and its transcript level is extremely high in the medulla oblongata and olfactory bulb (Figure 1B). Among six NKR, six NKR isoform could all be widely detected in various brain subregions and pituitary, and NK1Rb was more highly detected in brain compared to other NKRs (Figure 1B).” to “Finally, tissue distribution revealed that TAC4 was widely distributed in various brain regions except the cerebellum, and its transcript level was extremely high in the medulla oblongata and the olfactory bulb (Figure 1B). In addition, six NKRs could all be widely detected in various brain subregions and pituitary, and NK1Rb was more highly detected in brain compared to other NKRs (Figure 1B).”. (Line 179)

We have revised “To clarify the receptor selectivity of HK1 and HK2 for six NKR isoforms, six HEK-293T cell lines that could stably express NK1Ra, NK1Rb, NK2R, NK3Rb, NK3Ra1 or NK3Ra2 were established respectively and used for transfection study with nuclear factor of activated T-cells (NFAT) luciferace-expressing vectors,” to “To clarify the receptor selectivity of HK1 and HK2 for the six NKRs,  HEK-293T cell lines were established to stably express NK1Ra, NK1Rb, NK2R, NK3Rb, NK3Ra1 or NK3Ra2 were established, respectively. The cells were then used for transfection study with NFAT-Luc vectors,”. (Line 197)

We have revised “The results showed that the HK2 could activated all six NKR isoform, but HK1 displayed very weak activation for each NKR isoform (Figure 2)” to “The results showed that the HK2 could activated all 6 NKR isoforms, but HK1 displayed very weak activation for these NKRs”. (Line 201)

We have revised “Compared with the control group, 1051 differentially expressed genes (DEGs) were found in the HK2 treatment group,” to “Compared with the control group, a total of 1151 differentially expressed genes (DEGs) were found in the HK2 treatment group,”. (Line 217)

We have revised “In the cell components (CC) category, they were mostly enriched in plasma membrane, ion channel complexes and transcription factor complexes (Figure 3A). The results of KEGG enrichment analysis showed that the up-regulated DEGs werw mainly involved in the synthesis of steroid hormones, the cGMP-PKG signaling pathway and the Jak-STAT signaling pathway. The down-regulated DEGs is mostly concentrated in the cAMP signaling pathway,” to “The results of KEGG enrichment analysis showed that the up-regulated DEGs were mainly involved in the synthesis of steroid hormones, the cGMP-PKG signaling pathway and the Jak-STAT signaling pathway. The down-regulated DEGs were mostly enriched in the cAMP signaling pathway,”. (Line 232)

We have revised “However, HK1 (1μM) could not significantly regulated these genes in grass carp pituitary cells (Figure 5&6).,” to “However, HK1 (1 μM) showed no effect on these genes in grass carp pituitary cells (Figure 5&6).”. (Line 263)

We have revised “In parallel experiments to test the functional role of the PLC/IP3/PKC pathway,,” to “In parallel experiments for testing the functional role of the PLC/IP3/PKC pathway.”. (Line 307)

We have revised “. As shown in Figure 9A, our dose-response study had shown tht the HK1-V7F mutant could significantly enhance the stimulatory effects of HK1 on SLα,” to “As shown in Figure 9A, the dose-response study had showed that the HK1-V7F mutant could significantly enhance the stimulatory effects of HK1 on SLα,”. (Line 361)

We have revised “Similar to the results pituitary hormone genes regulation in carp pituitary cells, carp HK1with a “VXGLM” motif was found to be a relatively stimulant for six NKR isoforms” to “Similar to the results pituitary hormone genes regulation in carp pituitary cells, carp HK1 with a “VXGLM” motif was found to be a relatively low efficient stimulant for six NKR isoforms.”. (Line 367)

We have revised “Furthermore, we examined the antagonist effect of HK1 on HK2-induced pituitary hormone genes expression.” to “Furthermore, to verify whether HK1 could function as an endogenous antagonist of NKRs, we examined the antagonist effect of HK1 on HK2-induced pituitary hormone genes expression.”. (Line 374)

We have revised “have been reported” to “have been widely reported”. (Line 395)

We have revised “However, in the present study, only one TAC4 isoform was isolated in teleost, which might be due to the nonfunctionalization by forming pseudogenes or deletion/mutations leading to the loss of redundant genes” to “However, in the present study, only one TAC4 isoform was isolated in teleost, which might be the result of the non-functionalization by forming pseudogenes or deletion/mutations leading to the loss of redundant genes [24]”. (Line 398)

We have revised “In mice and rat, the HK-1 has similar affinity to SP at the NK1R [24-25], while human HK-1 binds to NK1R with very lower affinity than SP” to “Both in mice and rat, the HK-1 has similar affinity to SP at the NK1R [29-30], while human HK-1 binds to NK1R with very lower affinity than SP [31].”. (Line 412)

We have revised “we found six NKRs in grass carp, namely NK1Ra, NK1Rb, NK2R, NK3Ra1, NK3Ra2 and NK3Rb, resepectively. Basing on the ligand-receptor selecitivy, we found that HK2 could activate all six NKR isoforms, but display the highest affinity for NK2R.” to “6 NKRs were detected in grass carp, namely NK1Ra, NK1Rb, NK2R, NK3Ra1, NK3Ra2 and NK3Rb, respectively. Basing on the ligand-receptor selectivity, we found that HK2 could activate all 6 NKR isoforms, but show the highest affinity for NK2R..”. (Line 415)

We have revised “In mammals, many studies have focused on the actions of hemokinin on immunological regulation and inflammation” to “In mammals, many studies focused on the actions of hemokinin on immunological regulation and inflammation.”. (Line 424)

We have revised “In the present study, firstly, we found that six NKRs could be detected in grass carp pituitary, which indicated that HKs should play an important role in the pituitary.” to “In the present study, firstly, we found that six NKRs could be detected in grass carp pituitary, which indicated that HKs play an important role in the pituitary..”. (Line 426)

We have revised “In addition, HK1 could not activate any NKR isoforms in HEK-293T cells.” to “In addition, HK1 could barely activate any NKR isoforms in HEK-293T cells..”. (Line 455)

We have revised “In this case, the stimulatory effects of the HK1 mutant with the regeneration of the original “FXGLM” motif on SLα mRNA expression were markedly enhanced to the levels for HK2.” to “In this case, the stimulatory effects of the HK1-V7F with the regeneration of the original “FXGLM” motif on SLα mRNA expression was markedly enhanced to the levels for HK2.”. (Line 466)

Reviewer 2 Report

The manuscript entitled „Novel pituitary actions of TAC4 gene products in teleost” is well written and provide new insight into the TAC4 gene products actions in the teleost.

The abstract is informative. The introduction part briefly explains the scientific problem to be solved by the experiments. The methods are adequately applied for the scientific questions.

I detect only minor spelling mistakes which I indicated in the revised manuscript.

The manuscript is acceptable for publication after minor revision.

Author Response

Comments and Suggestions for Authors

The manuscript is acceptable for publication after minor revision.

Q1. I detect only minor spelling mistakes which I indicated in the revised manuscript.

Response:

Thank you very much for your help with the paper review along the way. As you suggested, we have tried our best to revise the manuscript again.

We have revised “luciferace” to “luciferase”. (Line 117)

We have revised “CO2” to “upper case for 2”. (Line 122)

We have revised “1μm” to “1μM”. (Line 132Line 224)

We have revised “Figureureure” to “Figure”. (Line 165)
